# Neighborhood Unsafety, Discrimination, and Food Insecurity among Nigerians Aged 15–49

**DOI:** 10.3390/ijerph20176624

**Published:** 2023-08-22

**Authors:** Chukwuemeka E. Ogbu, Chisa O. Oparanma, Stella C. Ogbu, Otobo I. Ujah, Ndugba S. Chinenye, Chidera P. Ogbu, Russell S. Kirby

**Affiliations:** 1Chiles Center, College of Public Health, University of South Florida, Tampa, FL 33612, USA; ogbu@usf.edu (C.E.O.); otobo@usf.edu (O.I.U.); 2Department of Medicine, Kharkiv National Medical University, 61022 Kharkiv, Ukraine; coparanma@gmail.com; 3Department of Biomedical Science, School of Medicine, Tulane University, New Orleans, LA 70112, USA; sogbu@tulane.edu; 4Department of Medical Sciences, University of Arizona, Tucson, AZ 85721, USA; chinaenyendugba@arizona.edu; 5Department of Biochemistry, Saint Joseph’s University, Philadelphia, PA 19074, USA; chidera.ogbu@sju.edu

**Keywords:** food insecurity, discrimination, neighborhood unsafety, gender differences, public health policy, Sub-Saharan Africa

## Abstract

We investigated the association between discrimination, neighborhood unsafety, and household food insecurity (FI) among Nigerian adults, as well as the gender-specific differences in these associations. Our analysis utilized data from the 2021 Multiple Indicator Cluster Survey (MICS), comprising 56,146 Nigerian adults aged 15–49 (17,346 males and 38,800 females). For bivariate analysis, we employed the Rao–Scott chi-square test to examine the relationship between predictors (discrimination, neighborhood unsafety, and a composite variable of both) and the outcome variable (FI). Food insecurity was assessed using both a dichotomous measure (food insecure vs. food secure) and a multinomial variable (food secure, mild FI, moderate FI, and severe FI). To model the association between predictors and FI while controlling for potential confounding factors, we utilized weighted binary and multinomial logistic regression. Among Nigerian adults, the prevalence of having ever experienced FI was 86.1%, with the prevalence of mild FI, moderate FI, and severe FI being 11.5%, 30.1%, and 44.5%, respectively. In the binary model, experiencing discrimination (OR = 1.36, 95% CI = 1.19–1.55), living in an unsafe neighborhood (OR = 1.33, 95% CI = 1.14–1.54), and facing both discrimination and unsafe neighborhood conditions (OR = 1.97, 95% CI = 1.57–2.48) were significantly associated with FI. In the multinomial model, discrimination, neighborhood unsafety, and experiencing both remained associated with moderate and severe FI. In the gender-specific models, discrimination and neighborhood unsafety were found to be significantly associated with FI in women but not in men. This study underscores the importance of implementing policies and programs that address the underlying causes of food insecurity, with specific attention to discrimination and neighborhood safety concerns, particularly for Nigerian women.

## 1. Introduction

Food insecurity, neighborhood unsafety, and discrimination are crucial factors affecting the well-being and health of individuals and communities [1]. These issues can exist independently, but their intersection often leads to complex challenges, especially for marginalized and vulnerable populations [1]. Food insecurity (FI) refers to the inadequate access to sufficient, nutritious food for maintaining an active and healthy life [2]. Globally, around 1.3 billion people suffer from food insecurity, with over half of them residing in Sub-Saharan Africa [2]. Notably, Nigeria faces a significant food insecurity crisis, with approximately 17 million people currently experiencing food insecurity and nearly 25 million Nigerians at risk of hunger by August 2023 [3]. In the 2022 Global Hunger Index, Nigeria ranked 103rd out of 121 countries, indicating a “serious” level of hunger [4]. Despite its efforts to improve food security, Nigeria’s Global Food Security Index (GFSI) ranking has shown an upward trend since 2013, but it still ranked 94th out of 113 countries in 2019 [5]. The North-East and North-Central states of Nigeria are particularly affected, where a combination of factors, including conflicts, insurgencies, kidnapping, armed banditry, cattle rustling, and extreme weather conditions, have exacerbated the situation [5]. Moreover, women bear a disproportionate burden, representing a significant portion of the world’s hungry population, with an estimated 60% of undernourished individuals being women [6]. Research has also indicated that women are more likely to experience food insecurity than men [7]. Therefore, addressing food insecurity is of paramount importance for public health and is deemed a national emergency in Nigeria.

The impact of food insecurity is complex, and one critical factor that significantly affects individuals’ ability to access and obtain food is neighborhood unsafety [8,9,10]. In neighborhoods with high crime rates and safety concerns, individuals may face barriers to purchasing food, resulting in limited access to affordable and nutritious options [9,10]. Moreover, residing in such areas can lead to reduced income due to factors like job loss, decreased economic activity, or increased medical expenses, making it challenging to afford healthy food choices [11]. The disruption of transportation and food distribution in insecure neighborhoods further exacerbates the problem, reducing food availability for both individuals and retailers and contributing to increased food insecurity [12]. Additionally, neighborhood unsafety takes a toll on mental health, causing stress and anxiety, which can hinder individuals’ capacity to access and prepare healthy food [13]. Particularly concerning is the impact on women, who are more vulnerable to violence and harassment in public spaces, limiting their access to food markets, social services, and other essential resources [14].

Discrimination, encompassing tribalism, sexism, xenophobia, ageism, ableism, religious discrimination, and other biases, profoundly impacts food insecurity [15]. Studies from various countries have shown that racial/ethnic disparities in health and social determinants, such as household food insecurity (HFI), can be linked to discrimination experienced in interpersonal and systemic contexts, disproportionately affecting daily functioning [15,16]. Discrimination curtails access to employment, education, healthcare, and resources, exacerbating food insecurity by reducing income and increasing the risk of poverty [17]. Moreover, discrimination’s effects on neighborhood safety are significant, as individuals facing discrimination may reside in marginalized communities with limited resources and higher crime rates [18]. This can result in social isolation, reduced access to support networks, and diminished mental well-being, further worsening the impact of food insecurity and neighborhood safety concerns [18]. Additionally, discrimination directly affects individuals’ physical and mental health, leading to chronic stress, trauma, and mental health disorders, further contributing to food insecurity and neighborhood safety issues [16,19,20].

In Nigeria, the vulnerable communities in the six regions of the country, which have been affected by terrorism, armed banditry, and kidnapping, experience a particularly noticeable intersectionality of food insecurity, discrimination, and neighborhood unsafety; however, there have not been studies that examined or explored these associations [3,5,21]. Studies have shown that communities facing food insecurity face systemic and structural barriers, including limited access to affordable and nutritious food, inadequate public infrastructure, and discriminatory policies and practices perpetuating food insecurity and neighborhood safety concerns [5,21,22]. For example, urban areas and low-income neighborhoods may lack grocery stores or have limited options for fresh produce, resulting in “food deserts” where residents have limited access to healthy food [23]. Marginalized communities may also face increased health disparities, including higher rates of diet-related diseases such as obesity, diabetes, cardiovascular diseases, and mental health issues related to chronic stress and trauma [17,23,24].

Given the disproportionate impact of food insecurity, neighborhood unsafety, and discrimination on women, it is essential to investigate gender differences in these associations [25,26]. However, there is a notable lack of research on this topic regarding Nigerian adults. Therefore, this study aims to address this gap and examine the following hypotheses:Discrimination and neighborhood unsafety independently and jointly predict food insecurity among Nigerian adults, when controlling for potential confounding factors.After considering confounders, there are gender disparities in the relationship between food insecurity, discrimination, and neighborhood unsafety.The association between food insecurity, discrimination, and neighborhood unsafety is more pronounced among individuals experiencing higher levels of food insecurity compared to those who are food secure, holding true for both genders within the adult population.

## 2. Materials and Methods

### 2.1. Study Design, Setting, Sampling Methodology, and Data Source

We utilized the 2021–2022 Multiple Indicator Cluster Survey-6 (MICS6), a cross-sectional study conducted among adults aged 15 to 49 in Nigeria [27]. Nigeria, the most populous African country, has an estimated population of approximately 221 million individuals, of whom 99.1 million are female [28]. In brief, MICS is a household survey program developed by the United Nations Children’s Fund (UNICEF) to collect data on the situation of children and women in low- and middle-income countries. MICS is designed to provide accurate and reliable information on various indicators related to child health, education, protection, and nutrition, maternal health and mortality, household environment, and water and sanitation [27]. The MICS dataset is publicly available to all interested researchers via the United Nations Children’s Fund MICS website (https://mics.unicef.org, accessed on 14 April 2023) upon request.

### 2.2. Study Sampling

MICS6 employed a stratified two-stage cluster sampling design, using enumeration areas (EAs), local government areas (LGAs), states, and zones in Nigeria, based on the 2006 Population Census. Enumeration areas (EAs) served as the primary sampling units (PSUs), established for the census enumeration. In the first stage, a list of households was compiled for each sample EA, and in the second stage, a sample of households was selected using systematic sampling or random selection. The survey achieved a response rate of 93.9% for men and 96.2% for women [27]. For more in-depth details on the sampling procedure, readers can refer to the MICS6 report [27]. In this study, however, we conducted a secondary data analysis using MICS6 data and utilized a convenience sample of adult individuals aged 15–49 years who responded to the Household Food Insecurity Questionnaire module. The total sample size comprised 56,146 participants (17,346 men and 38,800 women).

### 2.3. Ethical Approval

The MICS received approval from the Steering Committee and the Review Committee, which was formed by the UNICEF/Nigeria Technical Committee. Each respondent provided verbal consent prior to participation. Adult consent was obtained beforehand for children between the ages of 15 and 17 who were interviewed separately, followed by the child’s assent. All respondents were informed of the voluntary nature of their participation and the confidentiality and anonymity of their responses. Respondents were notified of their right to decline to answer any or all questions and discontinue the interview at any point. This study is exempt from full institutional review board review because the MICS dataset is publicly accessible.

### 2.4. Study Variables

#### 2.4.1. Outcome: Food Insecurity

The Food Insecurity Experience Scale (FIES) was employed to assess household food insecurity (FI) levels. Participants were asked, “During the last 12 months, was there a time when, because of lack of money or other resources: You were worried you would not have enough food to eat? You were unable to eat healthy and nutritious food? You ate only a few kinds of foods? You had to skip a meal? You ate less than you thought you should? Your household ran out of food? You were hungry but did not eat? You went without eating for a whole day?” Responses were recorded as yes (score = 1) or no (score = 0). The total score was used to categorize individual FI status into four groups based on the severity of their FI: food secure (score = 0), mild FI (scores = 1 to 3), moderate FI (scores = 4 to 6), and severe FI (scores = 7 to 8) [29]. In addition, a binary FI variable was derived by recoding the FIES scores into two categories, with a score = 0 indicating food-secure households and a score ≥1 indicating food-insecure households. The FIES is a well-validated metric of FI severity in Sub-Saharan Africa and relies on direct self-report of access to adequate food [30]. Each question in the FIES corresponds to a different experience and is linked to a particular degree of severity of food insecurity [31]. One of the distinguishing features of the FIES is its ability to include psychosocial aspects such as apprehension or uncertainty regarding access to sufficient food, in addition to assessing compromised dietary quality and reduced food intake; therefore, the FIES offers a reliable and valid tool for assessing FI in research contexts [30,31].

#### 2.4.2. Predictors: Neighborhood Unsafety and Discrimination

Neighborhood unsafety was assessed through two single-item questionnaire items that questioned participants’ perceptions of safety while walking alone in their neighborhood and safety at home after dark. Respondents who reported feeling “very safe” or “safe” were categorized as residing in a safe neighborhood, whereas those who indicated feeling “unsafe” or “very unsafe” were classified as living in an unsafe neighborhood. Questions regarding feelings of fear, perceptions of crime as a concern, and other aspects of perceived safety offer insight into individuals’ subjective sense of security in daily life. Such perceptions can significantly affect people’s mobility and decision-making regarding personal safety [32]. The questionnaire item “In last 12 months have you felt discriminated against or harassed on the basis of: Ethnic or immigration origin, gender, sexual orientation, age, religion or belief, disability, other/any reason” was used to assess individual discrimination. Any response indicating an affirmative experience was used to classify individuals as having ever experienced discrimination (yes vs. no).

#### 2.4.3. Confounders

The following confounders were selected based on a prior literature review [33,34]: age, educational status (classified as none/primary, secondary, or tertiary education), income quintiles (categorized as poorest, second, middle, fourth, and richest quintiles), urbanization status (dichotomized as rural or urban), household ownership of agricultural land (yes or no), household ownership of animal (yes or no), and geographic region (classified as South East, South South, South West, North Central, North East, or North West).

### 2.5. Data Analysis

MICS utilizes a stratified, multistage complex survey design to improve the representativeness of the Nigerian male population aged 15–49. Analytical guidelines that incorporate complex analytical procedures were followed as recommended. Weighting was applied to ensure the accuracy and reliability of the study findings. Descriptive statistics were calculated for all variables—means and standard deviations for continuous variables and frequencies and percentages for categorical variables. The Rao–Scott chi-square test (categorical) and t-test (continuous) were used for bivariate analysis between covariates and the outcome variables. Household FI was assessed as a dichotomous measure (food insecure vs. food secure) and a multinomial variable (food secure, mild FI, moderate FI, and severe FI). Both weighted binary and multinomial logistic regression were used to model the association between food insecurity and neighborhood unsafety while controlling for confounders. Similar regression analyses were used to model the association between food insecurity and having ever experienced discrimination, controlling for confounders. Odds ratios (ORs) and corresponding 95% confidence intervals (Cis) were reported. Weighted percentages were used to report all percentages. Multicollinearity was evaluated using the variance inflation factor (VIF), and no VIF value exceeded 10 [35]. Interaction terms between each predictor and confounders were tested and found statistically insignificant. Statistical significance was assessed using a two-sided *p*-value < 0.05 and nonoverlapping 95% Cis. Analysis was conducted using SAS 9.4 statistical software (SAS Institute, Cary, NC, USA).

## 3. Results

### 3.1. The Prevalence of Food Insecurity

The prevalence of having ever experienced FI (FIES score ≥ 1) was 86.1%. The prevalence of mild FI, moderate FI, and severe F1 were 11.5%, 30.1%, and 44.5%, respectively.

Compared to adults who did not experience discrimination, adults who had ever experienced discrimination (89.2% vs. 85.3%, *p* < 0.001) had a higher prevalence of food insecurity. Adults in unsafe neighborhoods (88.9 vs. 84.5%, *p* < 0.001) had a higher prevalence of food insecurity. Compared to adults who lived in safe neighborhoods and did not experience discrimination, adults living in unsafe neighborhoods and who reported discrimination had a higher prevalence of having ever reported food insecurity (92.3% vs. 83.9%, *p* < 0.05) (Table 1).

Among those who reported experiencing discrimination, the prevalence of moderate FI (30.8% vs. 29.9%) and severe FI (47.6% vs. 43.7%) was higher compared to those who did not report experiencing discrimination. Similarly, adults residing in unsafe neighborhoods had a higher prevalence of moderate FI (30.6% vs. 29.8%) and severe FI (47.2% vs. 43.0%) compared to those living in safe neighborhoods. Moreover, the prevalence of moderate FI (30.6% vs. 29.5%) and severe FI (51.7% vs. 42.6%) was higher among adults who reported both experiencing discrimination and living in unsafe neighborhoods compared to those who did not experience discrimination and lived in safe neighborhoods (Table 2).

### 3.2. The Association between Discrimination, Neighborhood Unsafety, and Food Insecurity

In the binary FI model, discrimination (OR = 1.36, 95% CI = 1.19–1.55), neighborhood unsafety (OR = 1.33, 95% CI = 1.14–1.54), and discrimination and neighborhood unsafety (OR = 1.97, 95% CI = 1.57–2.48) were associated with FI (Figure 1).

Multinomial logistic regression analysis models showed that experiencing discrimination was associated with moderate FI (OR = 1.37, 95% CI = 1.16–1.61) and severe FI (OR = 1.42, 95% CI = 1.24–1.63). Neighborhood unsafety was associated with moderate FI (OR = 1.29, 95% CI = 1.10–1.51) and severe FI (OR = 1.42, 95% CI = 1.21–1.66). Experiencing both discrimination and neighborhood unsafety was positively associated with mild FI (OR = 1.46, 95% CI = 1.02–2.10), moderate FI (OR = 1.89, 95% CI = 1.48–2.42), and severe FI (OR = 2.23, 95% CI = 1.76–2.82) (Figure 2).

### 3.3. The Association between Discrimination, Neighborhood Unsafety, and Food Insecurity by Gender

Discrimination (OR = 1.53, 95% CI = 1.27–1.84), neighborhood unsafety (OR = 1.39, 95% CI = 1.16–1.67), and experiencing both (OR = 2.09, 95% CI = 1.59–2.74) were significantly associated with FI in women in binary models. Either discrimination or neighborhood unsafety independently did not reach significance in men. However, men who experienced both discrimination and unsafety had 79% higher odds of reporting FI (OR = 1.79, 95% CI = 1.26–2.56) than men who did not experience both (Figure 1).

In the multinomial model, women who had ever experienced discrimination had higher odds of mild FI (OR = 1.20, 95% CI = 1.01–1.56), moderate FI (OR = 1.54, 95% CI = 1.24–1.90), and severe FI (OR = 1.61, 95% CI = 1.34–1.94). Neighborhood unsafety was associated with moderate FI (OR = 1.35, 95% CI = 1.11–1.64) and severe FI (OR = 1.50, 95% CI = 1.24–1.82) in women. Experiencing both discrimination and neighborhood unsafety was positively associated with moderate FI (women, OR = 2.00, 95% CI = 1.49–2.68; men, OR = 1.72, 95% CI = 1.16–2.55) and severe FI (men, OR = 1.94, 95% CI = 1.33–2.82; women, OR = 2.39, 95% CI = 1.82–3.14) in both genders. No significant association was observed between either discrimination or neighborhood unsafety and any of the FI severity classes in the multinomial model for men (Table 3).

## 4. Discussion

This study aimed to examine the association between food insecurity, discrimination, and neighborhood unsafety in Nigerian adults and the differences by gender.

Our study revealed that the prevalence of food insecurity (FI) (defined as FIES ≥ 1) among adult males and females aged 15–49 in Nigeria stands at 86%, while severe food insecurity (SFI) (defined as FIES 7–8) accounts for 44.5% of cases. These figures are situated within the prevalence ranges reported in other African countries. For instance, Wambogo et al. (2018), during the validation of the Food Insecurity Experience Scale for use in Sub-Saharan Africa and the characterization of food-insecure individuals, identified varied SFI prevalence rates across different regions of the continent [30]. In East Africa, SFI prevalence spanned from 20.7% in Ethiopia to a remarkable 87.3% in South Sudan. Central African nations such as Congo Kinshasa and Congo Brazzaville reported SFI prevalence rates of 40.4% and 48.4%, respectively [30]. For island nations like Mauritius and Madagascar, these rates were considerably lower, at 6.0% and 17.2%, respectively. In Southern Africa, Malawi exhibited the highest SFI prevalence at 64.5%, while South Africa reported the lowest at 19.0% [30]. Lastly, West Africa exhibited the widest range in SFI prevalence, from 6.6% in Mali to 71.5% in Liberia [30].

Our study also found that encountering discrimination, inhabiting neighborhoods perceived as unsafe, or experiencing both circumstances is associated with increased odds of FI within the adult population of Nigeria. The use of a multinomial model indicated that the occurrence of discrimination and residing in areas considered unsafe was associated with moderate and severe FI. This study corroborates earlier research that associated discrimination and unsafe neighborhoods with FI [36,37,38]. Odoms-Young and Bruce (2018) suggested that strategies addressing systemic discrimination may hold significant implications for diminishing racial/ethnic discrepancies in FI and fostering overall health equity [36]. Chung et al. (2012) concluded that neighborhood walkability is a critical associate of mobility-related food scarcity and worry regarding FI, even after adjusting for other pertinent variables [37]. Additionally, DiFiore et al. (2022) concluded that a lower prevalence of FI corresponded with higher perceived neighborhood safety and social cohesion [8]. The reasons why neighborhood unsafety and discrimination impact food insecurity are numerous but relate mainly to the accessibility and affordability of food. Individuals who are discriminated against often face limited access to employment opportunities, education, healthcare, and other resources, which can potentially diminish income and hamper their chances of procuring healthy and sustainable food options [17]. Discrimination may also indirectly influence neighborhood safety, given that those facing discrimination are more likely to reside in neighborhoods characterized by higher crime rates [18]. Unsafe neighborhoods deter individuals from procuring food from such locales [9,10], and neighborhoods characterized by high crime rates generally have diminished economic capability, making it challenging for residents to afford quality food [11].

When the data were stratified by gender, the association between FI and experiencing discrimination or living in unsafe neighborhoods was observed to be more pronounced and significant in women. This study further revealed that women residing in households severely or moderately affected by FI had increased chances of experiencing discrimination, living in unsafe neighborhoods, or encountering both circumstances compared to their counterparts in food-secure households. While the odds of encountering both predictors were significant in both genders, the odds ratio was higher in women. This suggests that gender may be a factor influencing the association between discrimination, neighborhood safety, and FI. Previous research by Botreau and Cohen (2020) also highlighted that discrimination and inequality lay at the core of FI, with women particularly susceptible to the pressure of providing wholesome meals and managing their children’s hunger [14]. This pressure, coupled with deeply rooted inequalities, hinder economic growth benefits from reaching women, who are typically marginalized socially, economically, and politically [38]. Quisumbing et al. (2011) found that women were more inclined to cut down their food consumption to ensure more availability for other family members [39]. Holmes et al. (2009) noted that women, despite receiving less remuneration than men for similar work, allocated a higher percentage of their income towards food than male heads of households [40].

Since its independence in 1960, Nigeria has implemented numerous programs and regulatory bodies to increase crop productivity, generate employment, and decrease food insecurity [41]. These include the Green Revolution, the Lower Niger River Basin Development Authority (LNRBDA), and Operation Feed the Nation (OFN). However, many programs are no longer functional due to weak institutional foundations and poor program implementation [41]. Newer programs were established in the hopes of tackling food insecurity. One such program is the National Food Security Program, which aims to ensure sustainable access, availability, and affordability of quality food to all Nigerians [42]. Action Against Hunger is a global nonprofit organization dedicated to combating hunger, and its focus is on addressing the fundamental causes of hunger by providing monthly food assistance to individuals in the northeastern states. They also support families in farming, livestock rearing, fishing, and other initiatives that enhance food accessibility for them. Despite implementing these strategies, these organizations often need help reaching underserved individuals residing in areas under the control of armed groups, which hinders their ability to provide the necessary support. Thus, improved efforts by the government to reduce conflict and improve neighborhood safety in these areas would allow a better reach of these resources to the marginalized communities that experience the most food insecurity.

Promoting women’s empowerment through education, skill-building, and income-generating activities can help reduce food insecurity by combating discrimination [14]. These programs can enhance women’s economic opportunities, decision-making power, and resource access, which will ultimately enable them to overcome financial inequalities and discrimination. Advocating for gender-responsive policies and legal reforms that address the structural causes of gender inequality is crucial. This may include promoting equal land rights, access to credit, and social protection measures that specifically consider the needs of women. Focusing on improving maternal and child nutrition can have long-term benefits for women’s food security as programs that provide nutrition education, prenatal care, breastfeeding support, and complementary feeding guidance can help address the nutritional needs of women and their children [42].

One limitation of this study is that it relied on self-reported measures of discrimination and neighborhood unsafety, which may be subject to social desirability or recall bias. Future studies should use objective measures of discrimination and neighborhood unsafety. More so, this study’s cross-sectional nature makes it challenging to determine whether discrimination and/or neighborhood unsafety directly led to food insecurity. Additionally, reverse causality might exist, where food insecurity leads to discrimination or perceptions of neighborhood unsafety. The unavailability of confounders in the dataset, like availability and quality of social support networks, mental health conditions (depression, anxiety, and post-traumatic stress disorder), and access to healthcare services, and possible residual confounders, can affect the association of discrimination or neighborhood unsafety and food insecurity. The strength of this study is that it uses large-scale data to enhance statistical power and increase the precision of the study’s estimates, allowing for more robust conclusions. Another strength is the sample representativeness of the Nigerian population and the external generalizability of our study findings, as well as using a validated measure of food insecurity in the country.

## 5. Conclusions

In conclusion, discrimination, neighborhood unsafety, and a composite variable of both is associated with food insecurity among Nigerian adults aged 15–49, with amplified impact of this association observed among women. While several initiatives have been undertaken by the Nigerian government to mitigate food insecurity, the persistence of structural impediments necessitates more concerted efforts, particularly in enhancing neighborhood safety and promoting gender-responsive policies. Future research should incorporate more objective measures of discrimination and consider additional potential confounders to further elucidate these relationships. Despite its limitations, this study provides a robust base, potentially guiding effective policy interventions to address food insecurity, specifically in a gendered context.

## Figures and Tables

**Figure 1 ijerph-20-06624-f001:**
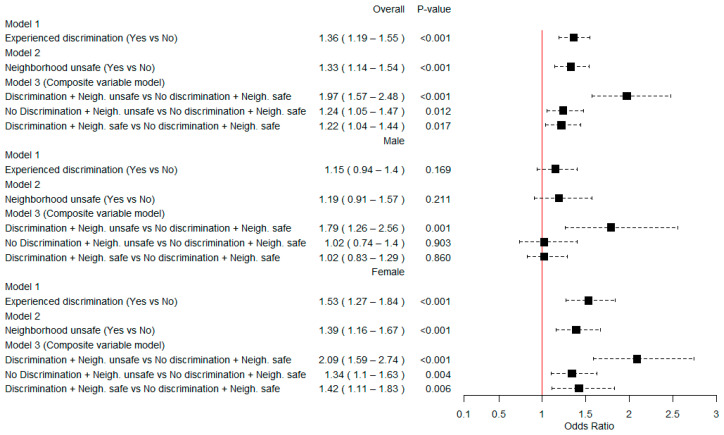
Overall and stratified weighted multivariable logistic regression of neighborhood unsafety, discrimination, and food insecurity among Nigerian adults aged 15–49 years; Multiple Indicator Cluster Survey, 2021. Models were adjusted for age, gender, educational level, marital status, wealth quintile, urbanization, region, household land ownership for agriculture, and household animal ownership in the overall model. Gender-specific models were adjusted for all covariates except for gender.

**Figure 2 ijerph-20-06624-f002:**
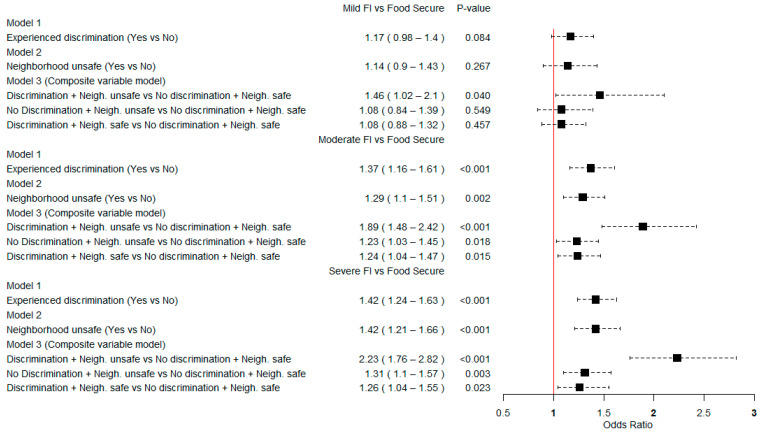
Weighted multinomial logistic regression of neighborhood unsafety, discrimination, and food insecurity among Nigerian adults aged 15–49 years; Multiple Indicator Cluster Survey, 2021.FI, food insecurity. Models were adjusted for age, gender, educational level, marital status, wealth quintile, urbanization, region, household land ownership for agriculture, and household animal ownership in the overall model.

**Table 1 ijerph-20-06624-t001:** The prevalence of food insecurity (binary) according to characteristics of Nigerian adults aged 15–49, the Multiple Indicator Cluster Survey, 2021.

	Total	Food Secure * (FIES Score = 0)	Food Insecure * (FIES Score ≥ 1)	*p*-Value
	No ^a^ (%) ^b^	No ^a^ (%) ^b^	No ^a^ (%) ^b^	
**Total**	56,146	7908 (13.9)	48,238 (86.1)	<0.0001
**Age (mean± SEM)**	28.7 (±10)	28.4 (±9.7)	28.7 (±10.0)	<0.0001
**Age categories**				0.60
15–19 years	13,257 (26.3)	1838 (13.7)	11,419 (86.3)	
20–29 years	17,917 (29.5)	2618 (14.0)	15,299 (86.0)	
30–39 years	14,020 (24.3)	2049 (14.3)	11,971 (85.7)	
40–49 years	10,952 (19.9)	1403 (13.4)	9549 (86.6)	
**Gender**				0.29
Male	17,346 (29.8)	2531 (14.3)	14,815 (85.7)	
Female	38,800 (70.1)	5377 (13.7)	33,423 (86.3)	
**Educational status**				<0.0001
None/primary school	7428 (12.8)	1656 (22.2)	5772 (77.8)	
Secondary school	25,515 (43.6)	3380 (13.8)	22,135 (86.2)	
Tertiary education	23,194 (43.6)	2869 (11.4)	20,325 (88.6)	
**Marital status**				0.26
No	22,294 (43.1)	3173 (14.3)	19,121 (85.7)	
Yes	33,852 (56.9)	4735 (13.5)	29,117 (86.5)	
**Wealth Quintile**				<0.0001
Poorest	12,590 (20.2)	1349 (9.1)	11,241 (90.9)	
Second quintile	12,700 (21.3)	1439 (10.6)	11,261(89.4)	
Middle	12,308 (20.6)	1277 (9.7)	11,031 (90.3)	
Fourth quintile	10,471 (19.6)	1437 (12.8)	9034 (87.2)	
Richest	8077 (18.3)	2406 (28.8)	5671 (71.2)	
**Urbanization**				<0.0001
Urban	17,979 (39.3)	3200 (17.1)	14,779 (82.9)	
Rural	38,167 (60.7)	4708 (11.7)	33,459 (88.3)	
**Household own agricultural land**				0.02
No	38,530 (66.7)	5096 (12.8)	33,434 (87.2)	
Yes	17,540 (33.3)	2788 (15.7)	14,752 (84.3)	
**Household own animal**				0.03
No	28,059 (51.9)	3874 (12.7)	24,185 (87.3)	
Yes	28,003 (48.1)	4029 (15.1)	23,974 (84.9)	
**Regions**				<0.0001
North Central	12,049 (14.7)	1528 (9.2)	10,521 (90.8)	
North East	11,922 (17.9)	2120 (18.2)	9802 (81.8)	
North West	12,404 (35.2)	1670 (12.4)	10,734 (87.6)	
South East	6580 (10.6)	486 (9.4)	6094 (90.6)	
South South	6635 (9.7)	653 (11.7)	5982 (88.3)	
South West	6556 (12.0)	1451 (22.9)	5105 (77.1)	
**Ever experienced discrimination**				<0.0001
No	43,012 (77.3)	6456 (14.7)	36,556 (85.3)	
Yes	13,116 (22.7)	1452 (10.8)	11,664 (89.2)	
**Neighborhood unsafety**				<0.0001
No	36,449 (63.0)	5619 (15.5)	30,830 (84.5)	
Yes	19,694 (37.0)	2289 (11.1)	17,405 (88.9)	
**Composite of discrimination and neighborhood unsafety**				<0.0001
Discrimination + unsafe neighborhood	5540 (9.7)	535 (7.7)	5005 (92.3)	
No discrimination + unsafe neighborhood	14,148 (27.3)	1754 (12.3)	12,394 (87.7)	
Discrimination + safe neighborhood	7576 (13.0)	917 (13.2)	6659 (86.8)	
No discrimination + safe neighborhood	28,862 (50.0)	4702 (16.1)	24,160 (83.9)	

FIES, Food Insecurity Experience Scale. * Food insecurity as a binary variable—food secure (FIES score = 0) and food insecure (FIES score ≥1). ^a^ Unweighted pooled sample size, 2021 MICS6. Due to item nonresponse, individual characteristic categories may not sum to total. ^b^ Weighted prevalence (%).

**Table 2 ijerph-20-06624-t002:** The prevalence of food insecurity (%) according to characteristics of Nigerian adults aged 15–49, the Multiple Indicator Cluster Survey, 2021.

	Total	Food Secure (FIES Score = 0)	Mild FI (FIES Score = 1–3)	Moderate FI (FIES Score = 4–6)	Severe FI (FIES Score = 7–8)	*p*-Value
	No ^a^ (%) ^b^	No ^a^ (%) ^b^	No ^a^ (%) ^b^	No ^a^ (%) ^b^	No ^a^ (%) ^b^	
**Total, N (%)**	56,146	7908 (13.9)	5911 (11.5)	16,705 (30.1)	25,622(44.5)	<0.0001
**Age (mean ± SEM)**	28.4 (±0.08)	28.4 (±0.20)	28.2 (±0.20)	28.7(±0.11)	28.4 (±0.10)	<0.0001
**Age categories**						0.37
15–19 years	13,257 (26.3)	1838 (13.7)	1393 (11.7)	3904 (29.7)	6122 (44.9)	
20–29 years	17,917 (29.5)	2618 (14.0)	1973 (11.9)	5274 (29.2)	8052 (44.9)	
30–39 years	14,020 (24.4)	2049 (14.3)	1438 (11.0)	4217 (31.1)	6316 (43.6)	
40–49 years	10,952 (19.9)	1403 (13.4)	1107 (11.3)	3310 (30.7)	5132 (44.7)	
**Gender**						0.48
Male	17,346 (29.8)	2531 (14.3)	1838 (11.6)	5190 (29.3)	7787 (44.9)	
Female	38,800 (70.2)	5377 (13.7)	4073 (11.5)	11,515 (30.4)	17,835 (44.4)	
**Educational status**						<0.0001
None/primary school	7428 (12.8)	1656 (22.2)	927 (14.6)	2083 (27.3)	2762 (36.0)	
Secondary school	25,515 (43.7)	3380 (13.8)	2570 (10.9)	7676 (30.2)	11,889 (45.1)	
Tertiary education	23,194 (43.6)	2869 (11.4)	2413 (11.3)	6944 (30.8)	10,968 (46.5)	
**Marital status**						0.08
No	22,294 (43.1)	3173 (14.3)	2259 (11.6)	6459 (28.8)	10,403 (45.3)	
Yes	33,852 (56.9)	4735 (13.5)	3652 (11.4)	10,246 (31.0)	15,219 (44.0)	
**Wealth Quintile**						<0.0001
Poorest	12,590 (20.2)	1349 (9.1)	1164 (9.5)	3655 (32.1)	6422 (49.4)	
Second quintile	12,700 (21.3)	1439 (10.6)	1353 (11.2)	3753(30.2)	6155 (48.0)	
Middle	12,308 (20.6)	1277 (9.7)	1243 (11.9)	3694 (27.6)	6094 (50.8)	
Fourth quintile	10,471 (19.6)	1437 (12.8)	1061 (10.6)	3279 (32.0)	4694 (44.5)	
Richest	8077 (18.3)	2406 (28.8)	1090 (14.6)	2324 (28.4)	2257 (28.2)	
**Urbanization**						0.0005
Urban	17,979 (39.3)	3200 (17.1)	1852 (11.7)	5271 (29.3)	7656 (41.9)	
Rural	38,167 (60.7)	4708 (11.7)	4059 (11.4)	11,434 (30.6)	17,966 (46.3)	
**Household own agricultural land**						0.07
No	38,530 (66.7)	5096 (12.8)	4165 (11.9)	11,534 (29.7)	17,735 (45.5)	
Yes	17,540 (33.3)	2788 (15.7)	1744 (10.7)	5156 (30.8)	7852 (42.7)	
**Household own animal**						0.02
No	28,059 (51.9)	3874 (12.7)	3228 (12.5)	8543 (31.3)	12,414 (43.5)	
Yes	28,003 (58.1)	4029 (15.1)	2673 (10.4)	8129 (28.8)	13,172 (45.7)	
**Regions**						<0.0001
North Central	12,049 (14.7)	1528 (9.2)	1083 (9.6)	3630 (29.8)	5808 (51.4)	
North East	11,922 (17.9)	2120 (18.2)	1369 (11.4)	3000 (25.8)	5433 (44.6)	
North West	12,404 (35.2)	1670 (12.4)	1569 (14.1)	4029 (31.8)	5136 (41.7)	
South East	6580 (10.6)	486 (9.4)	617 (9.5)	1937 (30.9)	3540 (50.1)	
South South	6635 (9.7)	653 (11.7)	585 (9.4)	2179 (32.5)	3218 (46.4)	
South West	6556 (11.9)	1451 (22.9)	688 (9.9)	1930 (29.1)	2487 (38.1)	
**Ever experienced discrimination**						<0.0001
No	43,012 (77.3)	6456 (14.7)	4662 (11.7)	12,676 (29.9)	19,218 (43.7)	
Yes	13,116 (22.7)	1452 (10.8)	1247 (10.7)	4026 (30.8)	6391 (47.6)	
**Neighborhood unsafety**						<0.0001
No	36,449 (63.0)	5619 (15.5)	3961 (11.7)	10,813 (29.8)	16,056 (43.0)	
Yes	19,694 (37.0)	2289 (11.1)	1950 (11.1)	5890 (30.6)	9565 (47.2)	
**Composite of discrimination and neighborhood unsafety**						<0.0001
Discrimination + unsafe neighborhood	5540 (9.7)	535 (7.7)	459 (10.1)	1631 (30.6)	2915 (51.7)	
No discrimination + unsafe neighborhood	14,148 (27.3)	1754 (12.3)	1491 (11.5)	4259 (30.6)	6644 (45.6)	
Discrimination + safe neighborhood	7576 (13.0)	917 (13.2)	788 (11.2)	2395 (30.9)	3476 (44.7)	
No discrimination + safe neighborhood	28,862 (50.0)	4702 (16.1)	3171 (11.8)	8416 (29.5)	12,573 (42.6)	

FI, food insecurity. ^a^ Unweighted pooled sample size, 2021 MICS6. Due to item nonresponse, individual characteristic categories may not sum to total. ^b^ Weighted prevalence in %.

**Table 3 ijerph-20-06624-t003:** Weighted multinomial logistic regression of neighborhood unsafety, discrimination, and food insecurity among Nigerian adults aged 15–49 years stratified by gender; Multiple Indicator Cluster Survey, 2021.

	Adjusted OR (95% CI)					
	Men			Women		
	Mild FI ^b^	Moderate FI ^b^	Severe F1 ^b^	Mild FI ^b^	Moderate FI ^b^	Severe FI ^b^
**Neighborhood unsafety** ^a^						
No	Ref	Ref	Ref	Ref	Ref	Ref
Yes	1.00 (0.62–1.62)	1.19 (0.90–1.57)	1.26 (0.94–1.71)	1.18 (0.92–1.51)	1.35 (1.11–1.64) ***	1.50 (1.24–1.82) ***
**Discrimination** ^a^						
No	Ref	Ref	Ref	Ref	Ref	Ref
Yes	1.03 (0.80–1.33)	1.15 (0.92–1.44)	1.19 (0.96–1.48)	1.20 (1.01–1.56) *	1.54 (1.24–1.90) ***	1.61 (1.34–1.94) ***
**Composite of discrimination and neighborhood unsafety** ^a^						
No discrimination + safe neighborhood	Ref	Ref	Ref	Ref	Ref	Ref
Discrimination + unsafe neighborhood	1.51 (0.86–2.67)	1.72 (1.16–2.55) **	1.94 (1.33–2.82) ***	1.48 (0.98–2.24)	2.00 (1.49–2.68) ***	2.39 (1.82–3.14) ***
No discrimination + unsafe neighborhood	0.79 (0.46–1.34)	1.05 (0.76–1.44)	1.08 (0.75–1.55)	1.18 (0.91–1.53)	1.31 (1.06–1.62) *	1.41 (1.15–1.73) **
Discrimination + safe neighborhood	0.89 (0.67–1.16)	1.06 (0.83–1.36)	1.08 (0.85–1.37)	1.33 (0.95–1.85)	1.48 (1.12–1.95) **	1.41 (1.09–1.83) **

CI, confidence interval; OR, odds ratio; FI, food insecurity. ^a^ Predictor adjusted for age, educational level, marital status, wealth quintile, urbanization, region, household land ownership for agriculture, and household animal ownership. ^b^ Comparator is food-secure category. * *p* < 0.05, ** *p* < 0.01, *** *p* < 0.001.

## Data Availability

The data used to generate the findings of this study are publicly available on the Multiple Indicator Cluster Survey Website available at: https://mics.unicef.org (accessed on 9 May 2023). The Multiple Indicator Cluster Survey (MICS) is a product of the United Nation’s Children Fund.

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
