# Peer review of "Neighborhood Unsafety, Discrimination, and Food Insecurity among Nigerians Aged 15–49"

_ijerph, 2023, doi:10.3390/ijerph20176624_

Round 1
Reviewer 1 Report
Ogbu et al. discovered that both discrimination and neighborhood unsafety were linked to moderate and severe food insecurity, particularly among women. This study holds valuable insights that can assist policymakers in enhancing food security in Nigeria. I have a few comments regarding the content:
1. It would be helpful if the authors compared their findings on overall high food insecurity levels with other studies and explained any discrepancies observed.
2. The structure and numbering of the methodology and results sections should be made clearer and more consistent.
3. To enhance accuracy, it would be preferable to present p-values rather than just the significance level. For example, adding an additional column of p-values to replace "c" in Tables 1 and 2, as well as replacing asterisks in Tables 3-5. Additionally, the table format could be improved by using three horizontal lines instead of the current format.
4. In Table 1, there is no difference in food insecurity between males and females, while in other results, women appear to be more vulnerable to food insecurity. It would be beneficial to include an explanation for this discrepancy.
5. It seems that the gender row is missing in Table 2. Please confirm if this is the case.
6. Is there any explanation for why higher education is associated with higher food insecurity?
7. While the study established the association between discrimination, neighborhood unsafety, and food insecurity, along with gender differences, the variations in food insecurity prevalence across factors are not dramatic (4%-10%). Given the persistently high overall prevalence of food insecurity, it would be interesting to investigate the relationship between additional factors and food insecurity prevalence, as this could help identify key determinants.
Author Response
Reviewer 1 comments:
Ogbu et al. discovered that both discrimination and neighborhood unsafety were linked to moderate and severe food insecurity, particularly among women. This study holds valuable insights that can assist policymakers in enhancing food security in Nigeria. I have a few comments regarding the content:
- It would be helpful if the authors compared their findings on overall high food insecurity levels with other studies and explained any discrepancies observed.
Thanks. This has been done and studies compared to other studies who used FIES. The paragraph reads:
“Our study revealed that the prevalence of food insecurity (FI) (defined as FIES ≥1) among adult males and females aged 15-49 in Nigeria stands at 86%, while severe food insecurity (SFI) (defined as FIES 7-8) accounts for 44.5% of cases. These figures are situated within the prevalence ranges reported in other African countries. For instance, Wambogo et al. (2018), during the validation of the Food Insecurity Experience Scale for use in Sub-Saharan Africa and the characterization of food-insecure individuals, identified varied SFI prevalence rates across different regions of the continent (31). In East Africa, SFI prevalence spanned from 20.7% in Ethiopia to a remarkable 87.3% in South Sudan. Central African nations such as Congo Kinshasa and Congo Brazzaville reported SFI prevalence rates of 40.4% and 48.4%, respectively (31). For island nations like Mauritius and Madagascar, these rates were considerably lower, at 6.0% and 17.2%, respectively. In Southern Africa, Malawi exhibited the highest SFI prevalence at 64.5%, while South Africa reported the lowest at 19.0% (31). Lastly, West Africa exhibited the widest range in SFI prevalence, from 6.6% in Mali to 71.5% in Liberia (31).
“
- The structure and numbering of the methodology and results sections should be made clearer and more consistent.
Thank you for your input on the structure and numbering of the methodology and results sections. We have now meticulously revised these sections for clarity and consistency. The methodology section has been divided into subsections, each denoted by a roman numeral. This format facilitates clearer and more organized presentation of the various aspects of our methodology. Similarly, the results section now mirrors this pattern, with each distinct set of findings grouped under a corresponding numeral. We believe this revision greatly improves the readability and coherence of our paper.
- To enhance accuracy, it would be preferable to present p-values rather than just the significance level. For example, adding an additional column of p-values to replace "c" in Tables 1 and 2, as well as replacing asterisks in Tables 3-5. Additionally, the table format could be improved by using three horizontal lines instead of the current format.
Response: Thank you for your valuable suggestions to improve the accuracy and presentation of our data. In response, we have updated Tables 1 and 2 to include p-values instead of the previous representation with the 'c' notation. As for Tables 3 and 4, we have replaced them with Forest Plots (Figures 1 and 2), using non-overlapping confidence intervals to indicate significance. Please note that any confidence interval that includes 1 should be considered not significant. We've followed the same approach for what is now Table 3 (previously Table 5). We believe these changes will enhance the interpretability of our results.
- In Table 1, there is no difference in food insecurity between males and females, while in other results, women appear to be more vulnerable to food insecurity. It would be beneficial to include an explanation for this discrepancy.
Response: Thank you for pointing out this. Specifically, Table 1 represents the proportions of food insecure versus food secure adults, broken down by gender, and does not show any significant difference between males and females. On the other hand, the other results highlight the associations between factors like discrimination and perceived neighborhood unsafety with food insecurity, and these associations were found to be more pronounced in women. This suggests that while the overall prevalence of food insecurity may not differ significantly by gender, the vulnerability to food insecurity in relation to discrimination and neighborhood unsafety might be higher among women.
- It seems that the gender row is missing in Table 2. Please confirm if this is the case.
Response: You are correct; the 'gender' row was inadvertently missing from Table 2. We have now included it, ensuring that the table is complete and accurate. We appreciate your vigilance in spotting this omission.
- Is there any explanation for why higher education is associated with higher food insecurity?
Response: Your question is insightful. While our study notes a correlation between higher education and food insecurity, it's crucial to emphasize that this does not signify a direct causative relationship. A few possible explanations could be higher levels of student loan debt among the educated, more expensive living conditions linked to their professional opportunities, or heightened awareness of dietary standards making them more cognizant of their food insecurity. However, these are merely speculations needing further research for validation. It's also important to clarify that our study's primary objective was not to identify factors driving food insecurity, but rather to investigate the associations between food insecurity, discrimination, and perceived neighborhood safety. Thus, 'education' was included as a confounding variable in our analysis, not as a predictor.
- While the study established the association between discrimination, neighborhood unsafety, and food insecurity, along with gender differences, the variations in food insecurity prevalence across factors are not dramatic (4%-10%). Given the persistently high overall prevalence of food insecurity, it would be interesting to investigate the relationship between additional factors and food insecurity prevalence, as this could help identify key determinants.
Response:
Thank you for your insightful comments and suggestions. We appreciate your observation regarding the relatively small variation in food insecurity prevalence across the factors we investigated (4%-10%). We agree that a broader investigation into additional factors contributing to food insecurity could potentially elucidate more dramatic variations and thereby identify key determinants. We have included this statement in the limitations of this study. While our current study aimed to elucidate the impact of discrimination, neighborhood unsafety, and gender differences on food insecurity, we acknowledge that these are not the sole contributors. Other factors such as income level, employment status, educational level, and access to social services can also play a significant role. Considering your valuable suggestion, we have revised our discussion section to acknowledge the potential influence of these additional factors and express the need for further research in these areas. We believe this amendment would help set the direction for future investigations aiming at addressing the persistently high overall prevalence of food insecurity. Thank you again for your thoughtful feedback.
Author Response
Review Report:
Thank you very much for providing me the opportunity to review this manuscript. While reading the manuscript, I could make the following comment from my side:
Summary: The current study was conducted to explore the relationship between discrimination, neighborhood unsafety, and household food insecurity (FI) among Nigerian adults, while also considering gender differences. The analysis involved data from the 2021 Multiple Indicator Cluster Survey (MICS), including 56,146 Nigerian adults (17,346 men and 38,800 women) aged 15–49. To examine the associations, the authors employed the Rao-Scott Chi-Square test for bivariate analysis, comparing predictors (discrimination, neighborhood unsafety, and a composite variable of both) with the outcome variable (FI). FI was assessed using two measures: a dichotomous measure (food insecure vs. food secure) and a multinomial variable (food secure, mild FI, moderate FI, severe FI). Weighted binary and multinomial logistic regression were utilized to control for confounders and model the association between predictors and FI. The findings revealed that, the prevalence of ever-experiencing FI among Nigerian adults was 86.1%. Among those affected, the prevalence of mild FI, moderate FI, and severe FI was 11.5%, 30.1%, and 44.5%, respectively. In the binary model, discrimination, neighborhood unsafety, and the combination of discrimination and neighborhood unsafety were associated with FI. Furthermore, in the multinomial model, discrimination, neighborhood unsafety, and experiencing both were linked to moderate and severe FI. However, when analyzing gender-specific models, discrimination and neighborhood unsafety showed significant associations with FI in women but were insignificant in men. These results emphasize the importance of implementing policies and programs that address the underlying causes of food insecurity, particularly discrimination and concerns regarding neighborhood safety. This is especially crucial for Nigerian women, as this study suggested that they are particularly affected by these factors.
Major Issues: The overall writing seemed good to me. The study revealed that, experiencing gender discrimination alone and living in an unsafe neighbourhood alone and also both of these factors have a strong association with experiencing various levels of food insecurities among adults in Nigerian communities.
Minor Issues:
This is a study with few writing errors/typos and I would suggest hiring a professional proof editor.
For instance, in the abstract, page 1, line 14, “Amount” should be changed to “among”. In the methodology section, it was mentioned that MICS6 data were utilized and a two staged stratified sampling techniques was used, again, participants were selected by convenient sampling. I was wondering whether the authors used only secondary data or again utilized the same PSUs based on the hypotheses of this current study. And if so, in an unstable environment like the study area (as discussed, the socioeconomic status of Nigeria), how did they manage to collect data for food insecurity, gender discrimination and neighborhood unsafety from a data set (MICS6) that was previously and primarily focused on maternal and child health. A more detailed and easy to understand explanation could be provided for readers and reviewers. Another thing that I noticed is that, there were two many tables, may be adding up a map or a figure would help the manuscript more presentable.
Response:
Thank you for taking the time to review our manuscript and for providing valuable feedback. We appreciate your thorough assessment, which has helped us identify areas for improvement in our study. We have carefully considered your comments and have taken the following actions to address the mentioned concerns:
Writing Errors and Typos:
We have conducted a comprehensive proofreading of the manuscript to correct all writing errors and typos. Specifically, we have rectified the mistake in the abstract on page 1, line 14, where "Amount" was changed to "among" to ensure accuracy and clarity.
Methodology and Sampling Techniques:
In response to the concern raised about our methodology, we acknowledge the utilization of MICS6 data and the two-staged stratified sampling technique with ultimate participant selection through convenient sampling. We have provided a detailed description of the survey methodology utilized by UNICEF's MICS, which collects household data through three datasets: men, women, and children. Our study utilized a convenient sample of all men and women aged 15-49 who participated in the MICS. To assess food insecurity and other variables, we adopted the same approach used by MICS, accessing household-level data. It is important to clarify that Nigeria is not deemed unstable. Discrimination, food insecurity, and neighborhood unsafety were among the variables collected, alongside others, as part of the MICS6 dataset.
This section now reads “MICS 6 employed a stratified two-stage cluster sampling design, using enumeration areas (EAs), local government areas (LGAs), states, and zones in Nigeria, based on the 2006 Population Census. Enumeration areas (EAs) served as the primary sampling units (PSUs), established for the census enumeration. In the first stage, a list of households was compiled for each sample EA, and in the second stage, a sample of households was selected using systematic sampling or random selection. The survey achieved a response rate of 93.9% for men and 96.2% for women [27]. For more in-depth details on the sampling procedure, readers can refer to the MICS6 report [27]. In this study, however, we conducted a secondary data analysis using MICS6 data and utilized a convenience sample of adult individuals aged 15-49 years who responded to the Household Food Insecurity Questionnaire module. The total sample size comprised 56,146 participants (17,346 men and 38,800 women).
“
Data Presentation:
We understand that the excessive number of tables may have made the manuscript less reader friendly. In response, we have reorganized the presentation of data and incorporated figures to improve the visual representation of our findings. This enhancement aims to make the manuscript more accessible and enhance the overall reading experience for the audience.
Once again, we sincerely appreciate your constructive feedback, which has undoubtedly improved the quality of our study. We are confident that the revised manuscript now addresses the concerns you raised, making it a more robust and impactful contribution to the field. We hope you find the changes satisfactory and would be grateful for any further.
Round 2
Reviewer 1 Report
Dear authors,
All comments were fully addressed.
Just move the titles below the figures. The period follows the figure instead of a colon.
Thanks!
Author Response
We have corrected the placement of figure titles as requested, and also corrected the x-axis scales in the figures.